# TROPICAL GEOMETRY FEATURES FOR NOVELTY DETECTION AND INTERPRETABILITY

## ABSTRACT

Existing methods for critical tasks such as out-of-distribution (OOD) detection, uncertainty quantification, and adversarial robustness often focus on measuring the output of the last or intermediate layers of a neural network such as logits and energy score. However, these methods typically overlook the geometric properties of the learned representations in the latent space, failing to capture important signals that relate to model reliability, fairness, and adversarial vulnerability.

Innovations: We introduce an innovative method, termed Tropical Geometry Features (TGF), for detecting out-of-distribution data and enhancing overall model evaluation. This approach leverages the geometric properties of polytopes derived from a trained neural network's learned representations. By integrating these geometric features with the data used during training, TGF establishes a unique signature of in-distribution data points. Our framework extends beyond OOD detection, providing insights into model uncertainty, adversarial robustness, interpretability, and fairness. Through TGF, we enhance interpretability techniques to detect OOD, uncertainty, adversarial robustness in dynamic and unpredictable environments.

## 1 INTRODUCTION

In the domain of machine learning (ML), the efficacy of a model is not solely measured by its predictive accuracy but also by its ability to provide reliable, interpretable, and fair outcomes under varying conditions. Beyond classification tasks, machine learning models must demonstrate versatility in tasks such as uncertainty quantification, out-of-distribution detection, adversarial robustness, fairness and bias detection, interpretability, and generalization to new tasks or domains (Mohseni et al., 2022; Amodei et al., 2016; Xu et al., 2018). These aspects ensure that the model not only works well on known data but can also be trusted and used effectively in dynamic, real-world environments.

For example, a well-trained classifier must not only provide high accuracy but also produce meaningful outputs that can be used for confidence calibration, enabling better uncertainty estimation (Guo et al., 2017), and for detecting adversarial examples, which are malicious inputs designed to fool the model (Carlini & Wagner, 2017). Additionally, the learned representations should support model interpretability, enabling humans to understand why a model made a particular prediction, and help in transfer learning, where features learned on one task can be reused on related tasks, thereby improving generalization (Yosinski et al., 2014).

While ML models have traditionally been developed with the assumption of closed-world scenarios, where test data is similar to the training data (Krizhevsky et al., 2012; He et al., 2015), the true utility of a model expands when it is capable of addressing diverse real-world challenges. For instance, models should be able to detect when they encounter out-of-distribution (OOD) data, handle adversarial threats, and ensure that their decisions remain fair and unbiased across different demographic groups. Furthermore, the model's internal representations should provide insights into its decision-making process, enhancing interpretability and explainability in complex systems.

In this work, we explore the theoretical and practical aspects of these learned representations, focusing on how neural networks manage uncertainty, detect OOD data, defend against adversarial inputs, and maintain fairness. Additionally, we build on the connection between neural networks and geometric structures, such as the relationship between feedforward neural networks with ReLU activation

functions and tropical geometry (Zhang et al., 2018). This geometric perspective offers a deeper understanding of how models represent data and adapt to different tasks and challenges, providing new avenues for enhancing the reliability, robustness, and interpretability of machine learning systems.

> *Can the tropical geometric features of the learned representations,*
> *provide insights into distinguishing in-distribution data from OOD data,*
> *assessing model uncertainty, detecting adversarial inputs,*
> *and improving interpretability and fairness?*

A neural network with a ReLU activation function can be decomposed into polytopes in the vector space where the data points reside, as discussed by Sudjianto et al. (2020). This property arises because the composition of linear functions with the ReLU activation function results in a piecewise linear function. Essentially, each piecewise linear segment corresponds to a distinct polytope in the vector space, delineating different regions where the neural network's behavior is linear.

The training process facilitates the learning of these polytopes, suggesting that the geometric attributes of the polytopes could be pivotal in delineating in-distribution (ID) data. We hypothesize that parameters such as the volume, the number of ID points contained within the polytopes, and their density could serve as indicators to characterize ID data effectively.

In fig. 1, we present a schematic representation of our proposed algorithm which we use to detect and explain out-of-distribution inputs to a pre-trained machine learning model. In the case that the input is tabular data and the machine learning model is a ReLU neural network, we analyse the entire network. In the case that the model expects some other form of data, e.g. a CNN for image detection, the model may consist of arbitrary initial layers that extract features (e.g. the flattened features extracted by a CNN), followed by ReLU feed-forward layers which we analyse. As ReLU feed-forward layers occur as a component of many types of networks, including transformers, there are many potential applications of our approach.

Traditional features for uncertainty quantification are based on just the final layer of the network. In many cases, the logits—i.e., the raw, unnormalized outputs of the neural network—are utilized as features for tasks such as out-of-distribution (OOD) detection or assessing adversarial vulnerability.

However, our approach diverges from this standard practice. We introduce novel features that incorporate a more comprehensive understanding of the ReLU network's structure and behavior, such as *volume* and *density* metrics. These features capture the complex interactions and characteristics of the entire ReLU network, rather than focusing solely on the final output layer. By considering the network holistically, our method aims to enhance the detection of OOD data, offer interpretations that may assist in understanding cases where inputs are misclassified (for example, due to outliers, dataset shift, or as a result of an adversarial attack) as well as provide additional characteristics to assist in detection of bias and vulnerabilities to adversarial attacks.

In this paper, we first define Tropical Geometry Features (TGF) as the set of geometric characteristics (e.g., volume, density) of the polytopes formed by the linear regions of a ReLU-activated neural network in the input space. We make our code publicly available for others that wish to experiment with our features[1] and outline the potential real-world applications of these features. Following this, we conduct an experiment to evaluate the suitability of TGF for OOD detection to see if our features are able to capture differences between in-distribution data and out-of-distribution data not detected by traditional logit-based features. Finally we conclude, noting that even weak performance on the task of detecting a single OOD point can lead to strong results on tasks such as dataset shift detection.

## 2 OUR ALGORITHM (**TGF**)

The family of feed forward neural network with ReLU activation functions is equivalent to the tropical rational maps Zhang et al. (2018). So if we are training a neural network with data coming from a $m$ dimensional space that would imply that we have a collection of polyhedra bounded by hyperplanes of dimension $m - 1$ in $\mathbb{R}^m$.

---

[1]Link to github repo removed for double-blind review

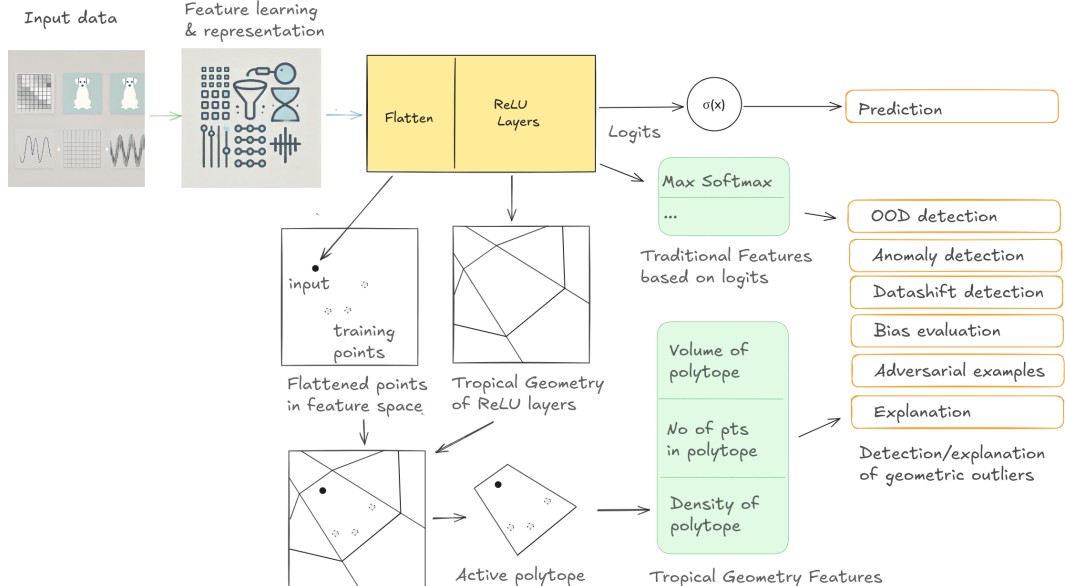

Figure 1: TGF for enhanced detection

Let us explain how this is done, it's a recursive process. First let's assume that the our training data is normalised, with norm less than one. Hence we can assume that all the data lives in the $m$ dimensional cube, which mathematically can be described by the intersection of the following hyperplanes $x_1 = \pm 1, x_2 = \pm 1, \ldots, x_m = \pm 1$.

Consider a feedforward neural network with $L$ hidden layers, employing the ReLU (Rectified Linear Unit) activation function:

$$\mathbb{R}^m \xrightarrow{(W_1, b_1)} \text{ReLU} \to \mathbb{R}^{h_1} \xrightarrow{(W_2, b_2)} \text{ReLU} \to \mathbb{R}^{h_2} \tag{1}$$

$$\to \cdots \to \mathbb{R}^{h_{L-1}} \xrightarrow{(W_L, b_L)} \text{ReLU} \to \mathbb{R}^{h_L} \xrightarrow{(W_{L+1}, b_{L+1})} \mathbb{R}^n. \tag{2}$$

In this configuration:

- $\mathbb{R}^m$ represents the input space.
- $\mathbb{R}^n$ denotes the output space.
- $h_i$ signifies the number of nodes in layer $i$.
- Layer 0 corresponds to the input space ($h_0 = m$) and layer $L + 1$ to the output space ($h_{L+1} = n$).
- $W_i \in \mathbb{R}^{h_i \times h_{i-1}}$ and $b_i \in \mathbb{R}^{h_i}$ are the weight matrix and bias vector for layer $i$, respectively.

The activation functions for hidden layers (layers 1 through $L$) are ReLU functions, applied coordinate-wise. The transition to the last layer (output layer) is an affine linear map without the application of a ReLU function.

The ReLU function is defined as:

$$\text{ReLU}(a) = \begin{cases} a & \text{if } a > 0, \\ 0 & \text{if } a \leq 0. \end{cases} \tag{3}$$

This function is a piecewise linear and continuous map on real numbers. For a vector $x \in \mathbb{R}^{h_i}$, the ReLU function is applied to each coordinate.

Let $w_{i,j}$ represent the $j$th row of the matrix $W_i$ and $b_{i,j}$ the $j$th entry of $b_i$. For an input data point $x \in \mathbb{R}^m$, the output in layer $i$ is denoted by $F_i(x)$. Thus, with this notation we have $F_i(x) \in \mathbb{R}^{h_i}$, $F_0(x) = x$ and

$$F_i(x) = \text{ReLU}(W_i F_{i-1}(x) + b_i) =$$

$$\begin{bmatrix} \max\{0, w_{i,1} F_{i-1}(x) + b_{i,1}\} \\ \vdots \\ \max\{0, w_{i,h_i} F_{i-1}(x) + b_{i,h_i}\} \end{bmatrix}$$

The composition of linear maps remains linear, the composition of a linear map with the ReLU function results in a function that is stepwise linear. This property is crucial in analyzing the geometric transformations performed by layers in neural networks, especially in understanding how data is transformed as it passes through the network.

Let $f : \mathbb{R}^m \to \mathbb{R}^p$ and $g : \mathbb{R}^p \to \mathbb{R}^q$ be linear maps. These maps satisfy additivity $f(x + y) = f(x) + f(y)$ and homogeneity $f(\alpha x) = \alpha f(x)$ for $x, y \in \mathbb{R}^m$, $\alpha \in \mathbb{R}$. Their composition $g \circ f$ is also linear.

The ReLU function, defined as $\text{ReLU}(x) = \max(0, x)$, is piecewise linear but not globally linear due to its behavior at $x = 0$.

In a neural network, a layer's transformation $R \circ L$ where $L : \mathbb{R}^m \to \mathbb{R}^p$ is linear and $R : \mathbb{R}^p \to \mathbb{R}^p$ is ReLU, is stepwise linear. This implies that $R \circ L$ is linear within certain intervals of $\mathbb{R}^m$.

This property allows hyperplanes in $\mathbb{R}^p$ to be mapped back to $\mathbb{R}^m$ in a piecewise linear manner, a feature unique to ReLU and not necessarily applicable to other activation functions.

## 2.1 Devision of space using hyperplanes corresponding to ReLU activation point

A hyperplane in an $m$-dimensional vector space $\mathbb{R}^m$ is a subspace that has dimension $m - 1$. It can be described as the set of vectors that satisfy a linear equation of the form:

$$ax_1 + bx_2 + \cdots + kx_m = d \tag{4}$$

where $a, b, \ldots, k$ and $d$ are constants in the field $\mathbb{R}$ (the real numbers), and at least one of $a, b, \ldots, k$ is non-zero. Here, $x_1, x_2, \ldots, x_m$ are the coordinates of a vector in $\mathbb{R}^m$.

In terms of a linear map, a hyperplane in $\mathbb{R}^m$ can be interpreted as the kernel (or null space) of a non-zero linear function. A linear functional $f : \mathbb{R}^m \to \mathbb{R}$ maps vectors in $\mathbb{R}^m$ to scalars in $\mathbb{R}$. The kernel of $f$, denoted as $\ker(f)$, is the set of all vectors $v \in \mathbb{R}^m$ such that $f(v) = 0$. When $f$ is non-zero, $\ker(f)$ is a hyperplane in $\mathbb{R}^m$.

Thus, the hyperplane can be formally defined as:

$$\text{Hyperplane} = \{v \in \mathbb{R}^m \mid f(v) = d\}$$

where $f : \mathbb{R}^m \to \mathbb{R}$ is a non-zero linear functional and $d$ is a constant in $\mathbb{R}$.

In this generalized setting of a linear map $f : \mathbb{R}^m \to \mathbb{R}^n$, each component function of $f$, considering $f$ as a vector of $n$ scalar-valued functions $f = (f_1, f_2, \ldots, f_n)$, can define a hyperplane in $\mathbb{R}^m$. For each linear functional $f_i : \mathbb{R}^m \to \mathbb{R}$, a hyperplane can be defined as:

$$\{v \in \mathbb{R}^m \mid f_i(v) = d_i\}$$

for some constant $d_i$.

## 2.2 Polytopes from Linear Maps

A polytope in $\mathbb{R}^m$ can be defined as the set of points satisfying a system of linear inequalities represented by $f$. These inequalities define a region in $\mathbb{R}^m$ bounded by hyperplanes. Formally, a polytope $P$ can be defined as:

$$P = \{x \in \mathbb{R}^m \mid Ax \le b\}$$

where $A$ is an $n \times m$ matrix representing the linear map, and $b$ is a vector in $\mathbb{R}^n$. Each row of $A$ and the corresponding element of $b$ define one of the bounding hyperplanes of the polytope.

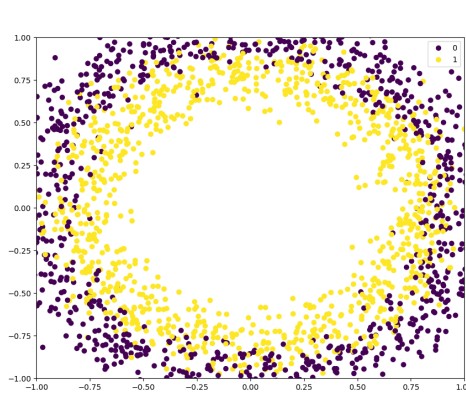 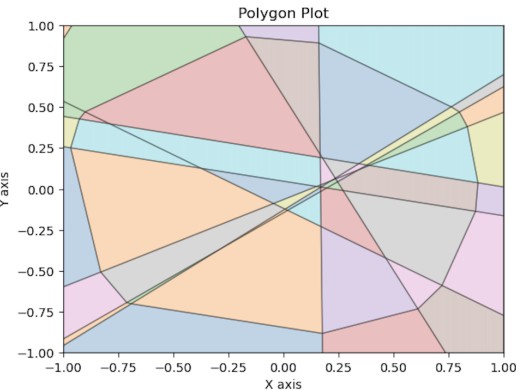

(a) Noisy points around a circle

(b) Neural network classification regions after training

Figure 2: Polygonal decomposition for a neural network trained to classify noisy points lying around a circle.

## 2.3 POLYTOPE DECOMPOSITION OF A NEURAL NETWORK

**Overview** In this subsection, we examine the decomposition of polytopes derived from a ReLU neural network, as detailed in our neural network model. The analysis focuses on the transformation of input space through each layer of the network, leading to a piecewise linear mapping that can be conceptualized as a collection of hyperplanes in $\mathbb{R}^m$.

**First Layer Transformation** Consider the first transformation in the network:

$$\mathbb{R}^m \xrightarrow{(W_1, b_1)} \text{ReLU} \to \mathbb{R}^{h_1}$$

As ReLU is piecewise linear, this transformation results a piecewise linear map from $\mathbb{R}^m$ to $\mathbb{R}^{h_1}$. Each linear region in this map is a polytope bounded by hyperplanes corresponding to the non-linearity in the ReLU activation at $a = 0$ for each of the hidden neurons.

**Recursive Process through Layers** The process extends recursively to subsequent layers. For instance, consider the second layer, which involves the following composite mapping:

$$\mathbb{R}^m \xrightarrow{(W_1, b_1)} \text{ReLU} \to \mathbb{R}^{h_1} \xrightarrow{(W_2, b_2)} \text{ReLU} \to \mathbb{R}^{h_2}$$

As the first layer transformation results in a piecewise linear map composed of linear regions in $\mathbb{R}^m$, we can apply the process recursively to each of these linear regions. The final result is therefore another piecewise linear map from $\mathbb{R}^m$ to $\mathbb{R}^{h_2}$

**Hyperplane-Induced Decomposition** Each hyperplane effectively divides $\mathbb{R}^m$ into two distinct regions:

- The region where $ax_1 + bx_2 + \cdots + kx_m - d > 0$ is assigned a binary value of 1.
- The region where $ax_1 + bx_2 + \cdots + kx_m - d < 0$ is assigned a binary value of 0.

**Classification** Without loss of generality, in this paper we examine neural networks designed to predict a probability (e.g. the probability that an image is of a cat); as such the final output passes though a sigmoid function to transform a logit of range $\mathbb{R}$ to a probability within the range [0, 1]. Our approach allows for optionally including one final decomposition of positive logits (i.e. corresponding to probabilities > 0.5) from negative logits (corresponding to probabilities < 0.5). However, our approach does not allow for hidden sigmoid layers as unlike ReLU layers, sigmoid operations cannot be composed as a piecewise linear map.

In fig. 2, we demonstrate how a two-dimensional dataset transforms into a collection of two-dimensional polytopes. fig. 2(a), represents the two-dimensional dataset. After training the neural network to classify the points, the resulting polygonal decomposition is illustrated in fig. 2(b). These polygons are two-dimensional because the data itself is two-dimensional. Each polygon is assigned a unique label, generated recursively through the neural network layers by binary combinations of "0" for inactive neurons and "1" for active neurons.

### 2.4 FEATURES OF POLYTOPES

In the section 2.3 we discussed how we obtained the polytopes from trained neural networks. We now assign the following features to the polytopes.

**Binary vector assignment to polytopes**   Each polytope, defined by the bounds of these hyperplanes, can be associated with a binary vector of dimension $h_1 + \cdots + h_L$. Each coordinate in this vector corresponds to a neuron in the network, the active is represented by $> 0$ and inactive by $< 0$. This representation allows for a clear and structured understanding of how the input space is transformed and partitioned across the layers of the neural network.

**Volume assignment to polytope**   There are various mathematical ways we can define the volume of polytopes such as the triangulation method, divergence theorem, Monte Carlo method etc. Let's choose the triangulation method along with the determinant method for calculating the volume.

Consider a convex polytope $P$ in $\mathbb{R}^m$ defined as:

$$P = \{x \in \mathbb{R}^m \mid Ax \leq b\}$$

where $A$ is an $n \times m$ matrix and $b$ is a vector in $\mathbb{R}^n$. Each row of $A$ along with the corresponding element of $b$ defines a hyperplane, and the polytope $P$ is the intersection of the half-spaces defined by these inequalities.

To compute the volume of $P$, one method is to decompose the polytope into simplices and calculate the volume of each simplex. For a simplex in $\mathbb{R}^m$ defined by vertices $v_0, v_1, \ldots, v_m$, the volume is given by:

$$\text{Volume of simplex} = \frac{1}{m!} \left| \det([v_1 - v_0, v_2 - v_0, \ldots, v_m - v_0]) \right|$$

If the polytope $P$ is decomposed into $k$ simplices $S_1, S_2, \ldots, S_k$, then the volume of $P$ is the sum of the volumes of these simplices:

$$\text{Volume of } P = \sum_{i=1}^{k} \text{Volume of } S_i$$

This method requires an effective decomposition of $P$ into simplices, which can be complex, especially in higher dimensions.

**Density assignment to polytope**   Consider the training of a Neural Network using in-distribution data. Let the polytope formed in the feature space of the network be denoted as $P$. The number of in-distribution data points within polytope $P$ is denoted by $|P|$, representing the count of training data points that fall inside $P$.

The volume of the polytope $P$, denoted as $\text{Vol}(P)$, represents the size of $P$ in the feature space. This volume can be calculated using tools described in the previous section.

The density of in-distribution data points within $P$ is defined as the ratio of the number of data points in $P$ to the volume of $P$. Mathematically, this is expressed as:

$$\textbf{Density}(P) = \frac{|P|}{\text{Vol}(P)} \tag{5}$$

This formulation of **Density**$(P)$ provides a metric for evaluating how densely the in-distribution data points are populated within the polytope $P$. A higher value of **Density**$(P)$ indicates a region with a

higher concentration of data points, while a lower value suggests a sparser region. Unlike traditional density-based approaches which require defining bounds or parameters to determine the regions over which density is calculated, our technique is fundamentally different. Here, the regions are derived directly from the structure of the neural network.

# 3 USE CASE OF TGF IN MACHINE LEARNING MODELS

In this section we outline potential applications of TGF. In our experiment we focus on OOD detection, noting that if TGF can provide an improvement over a traditional OOD baseline, such features are likely to also find applications in many other use cases outside of just OOD. We leave it to the community to experiment and find more applications of the **TGF**.

- **Model Reliability and Robustness Techniques** Uncertainty Quantification, Out-of-Distribution (OOD) Detection, adversarial Example Detection, and Network Robustness Analysis are techniques closely related to the evaluation of model reliability and robustness. For instance, in traditional OOD detection, the logits, or the output from the final fully connected layer of a neural network, are typically employed to identify whether an input lies outside the distribution of the training data Djurisic et al. (2022); Wang et al. (2022). In this work, we demonstrate how our Tropical Geometry Features (TGF) approach can be effectively utilized for OOD detection. Furthermore, we argue that TGF has potential applications in other related areas, such as uncertainty quantification and adversarial robustness.

- **Model Interpretability** The paper by Geva et al. (Geva et al., 2020) reveals that feed-forward layers in transformer models store and utilize patterns from the training data to make predictions about subsequent words. Simpler patterns are captured in the lower layers, while more complex patterns are learned in the upper layers. This insight enhances our understanding of the role these layers play in the overall language comprehension of transformer models. A crucial aspect of the feed-forward layers is that their activation function is ReLU, which, when considered within the **TGF** framework, suggests a potential for exploring deeper connections. Other works such as Pollano et al. (2023), can be seen through the lens of TGF.

- **Bias Evaluation** In regions that are data rich, neural networks are able to learn a detailed approximation of how inputs map to the predicted output, which may require many polytopes. However, in regions that are data poor, we would expect a course approximation. Therefore, in cases where the distribution of tropical geometry features differs between groups, e.g. groups based on age, race, or gender, this may be an indicator that the neural network does not perform equally well on all groups, most likely as a result of insufficient training data for certain groups.

  From an implementation standpoint, this is similar to out of distribution detection, but in the case of protected attributes should be interpreted as a failure of the network to behave appropriately for that input rather than an issue with the input.

- **adversarial attack** adversarial attacks involve subtly manipulating the input data of a classifier, such that models like deep neural networks misclassify the altered input with high confidence, even though the changes are almost imperceptible. This vulnerability raises serious concerns about the reliability of these models, particularly in safety-critical applications Dalvi et al. (2004); Lowd & Meek (2005). Although several countermeasures have been proposed Gu & Rigazio (2014); Goodfellow et al. (2014), these methods exhibit significant limitations. In their work, Hein et al. Hein & Andriushchenko (2017) investigated the extent to which a classifier's decision remains stable (i.e., does not change) when the input is perturbed within a certain radius in the input space. We anticipate that our proposed **TGF** approach could enhance robustness against adversarial attacks, as it incorporates additional information about the data residing within specific polytopes and leverages its geometric features.

- **Dataset Shift** Even weak evidence that a data point is out-of-distribution or anomalous can become strong evidence when we observe many such values. Thus a practical application of our approach could be early alerts of potential dataset shift when the distribution of volume or density features differs to that of the training data.

# 4 EVALUATION OF TGF FOR OOD DETECTION

In this section, we conduct an experiment to answer the following questions:

- Can the feature volume, which does not require access to training data points, be used as a standalone feature for better OOD detection? How does it compare to existing baseline methods for OOD detection without access to training data?

- Can TGF features that combine structure with knowledge of the training distribution, such as the TGF density feature, result in a better OOD detection?

## 4.1 EQUIVALENCE OF BASELINE OOD METHODS WHEN APPLIED TO BINARY CLASSIFICATION

Existing baseline techniques for OOD detection include maximum softmax probability Hendrycks & Gimpel (2017) and energy score Liu et al. (2020). These are computed from the logits for each class in the last layer of a neural network. While these techniques differ in performance for multi-class prediction, in the case of a neural network that performs binary classification, the output may consist of only a single logit that is passed through a sigmoid function to produce a probability between 0 and 1. Any monotonic transformation will have no effect on the separability of in-distribution data from OOD data.

As such, the baseline method used in this paper for OOD detection is the absolute value of the logit score, which can be thought of as a measure of how confident the neural network is in its prediction. The absolute value is taken as negative values of the logit corresponding to probabilities below 0.5 can be interpreted as confidence in the negative class. For ease of visualising large values, we further log transform the result when plotting distributions, noting that monotonic transformations, such as log, exp, and sigmoid, have no effect on the separability of in-distribution data from OOD data.

## 4.2 DATASETS

It is known that the performance of any given OOD method is highly dependent on the dataset Tajwar et al. (2021). As such, we test on multiple datasets as well as different types of OOD data.

For this experiment, we use three different in-distribution datasets: Higgs (a physics dataset, which we reduce to 6 input dimensions), Skin (a simple classification challenge with 3 input dimensions representing a color), and Circle (a synthetic dataset with 2 dimensions). Details of each dataset are provided in appendix A.1. We performed a test/train split, in which training data was used for model training as well as reporting the number of training points in each polytope. The in-distribution test data was used for computing TGF, as well as for generating OOD datasets that distort the test data in some manner. The full list of methods we use for generating OOD data is described in appendix A.2 and the number of test/train points for each dataset are listed in appendix B.

## 4.3 MODELS

For each of the datasets, we trained a fully connected ReLU network to predict the output. These consisted of one or more hidden ReLU layers followed by a sigmoid function in the final output layer to convert the result to a probability between 0 and 1. Details of the structure of each network is provided in appendix B. Note that our goal is not to train a state of the art neural network, but rather to provide a sample of ReLU neural networks trained for different datasets on which to test TGF.

## 4.4 METHODOLOGY

We use our TGF framework, described in section 2, to construct the polytopes from the neural network. Each polytope is assigned a unique label. We calculate the number of points, volume, and density for each polytope.

For each in-distribution test point that lies within a polytope, we attach the features of volume and density of the polytope to it. This process allows us to obtain a distribution for each feature. We repeat the same process for each of the OOD test points. We evaluate the ability of each TGF feature to separate an OOD data point from the in-distribution data. We report the False Positive Rate at 95%

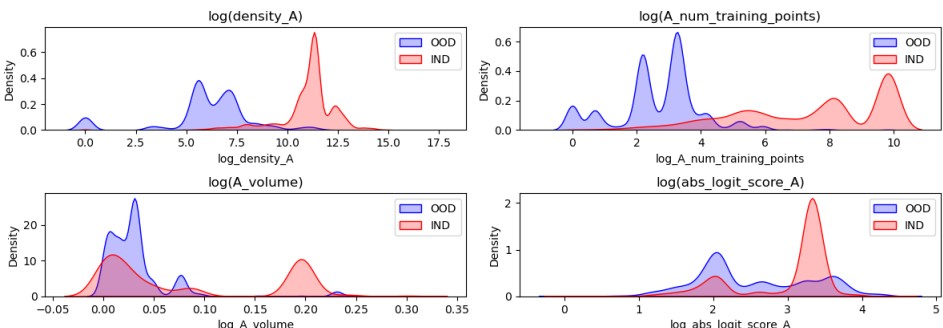

Figure 3: Comparative Analysis of Various Attributes for ID and OOD (color inversion method described in appendix item 2) for the Skin dataset

(FPR@95), a widely-used metric in OOD detection. It represents the percentage of OOD examples that are incorrectly classified as in-distribution (false positives), when the classification threshold is set such that 95% of in-distribution examples are correctly classified (true positives).

In the case of baseline methods based on logits, it is assumed that a larger absolute logit (higher confidence) is more likely to be in-distribution, while points with lower confidence are out-of-distribution. For TGF, we hypothesise that in-distribution data points are more likely to fall in high density polytopes (small volume with many training points) while OOD data will fall in low density polytopes (large volume with few training points). A random classifier will obtain a FPR@95% score of 0.95, a false positive rate above this indicates the assumptions were incorrect.

## 4.5 RESULTS

The results of our experiment are presented in tables 1 to 3. To provide more insight into the ability of different features to distinguish in-distribution data from out-of-distribution data, we show the feature distributions for the the Skin dataset in fig. 3. We also include the feature distribution for the Higgs boson dataset in the Appendix (fig. 4).

The OOD dataset created using **OOD_permu** is particularly challenging to detect, as the data in each column retains the same distribution, despite being permuted to create the OOD samples. We demonstrate that our feature **density** continues to perform excellently.

Our **volume** feature was worse than random (i.e. FPR@95% > 0.95) on the Higgs dataset. Nevertheless, we can see that for Skin and Circle datasets there were certain types of OOD data where **volume** performed better than random while **logits** performed worse than random. This suggests that despite being a weak classifier, it is extracting features that a baseline classifier based on logits doesn't, opening up the possibility of a hybrid approach.

| Features | FPR@95% for OOD classification (lower is better) | | | |
|---|---|---|---|---|
| | **OOD_permu** | **OOD_noise** | **OOD_nonlinear** | **OOD_rearrange** |
| Logits | 0.98 | 0.98 | 0.97 | 0.97 |
| Volume | 0.97 | 0.98 | 0.99 | 0.99 |
| No. of Training Points | 0.47 | 0.06 | 0.062 | 0.062 |
| Density | **0.44** | **0.03** | **0.025** | **0.025** |

Table 1: False Positive Rate at 95% (lower is better) for different features on the Higgs dataset across OOD datasets.

| Features | FPR@95% for OOD classification (lower is better) | | | | |
|---|---|---|---|---|---|
| | OOD_min | OOD_inv | OOD_diff_dist | Textures | OOD_permu |
| Logits | 0.84 | **0.39** | 0.99 | 0.88 | 0.96 |
| Volume | 0.99 | 0.92 | 0.99 | 0.77 | 0.54 |
| No. of Training Points | 0.70 | 0.49 | 0.69 | 0.50 | 0.87 |
| Density | **0.57** | 0.57 | **0.59** | **0.11** | **0.26** |

Table 2: False Positive Rate at 95% (lower is better) for different features on the Skin dataset across OOD datasets.

| Features | FPR@95% for OOD classification (lower is better) | | | | |
|---|---|---|---|---|---|
| | OOD_min | OOD_noise | OOD_nonlin | OOD_max | OOD_permu |
| Logits | 0.99 | 0.96 | 0.98 | 0.96 | 0.95 |
| Volume | 0.93 | 0.92 | 0.90 | 0.96 | 0.83 |
| No. of Training Points | 0.79 | 0.84 | 0.87 | 0.77 | 0.88 |
| Density | **0.73** | **0.78** | **0.85** | **0.72** | **0.52** |

Table 3: False Positive Rate at 95% (lower is better) for different features on the Circle dataset across OOD datasets.

## 5 DISCUSSION

A key benefit of our proposed volume feature is that it can be computed from the model weights without requiring any access to the training data. This makes the approach well suited for applications where the training dataset cannot be shared for privacy or commercial reasons, for example, a classifier trained on sensitive information. While the false positive rate was too high to safe-guard against single OOD points, as argued in section 3 even a weak OOD detector for a single data point can be a strong detector for other applications. In particular, we showed that the features it extracts differe to baseline approaches based on logits, performing well in some cases where logits fails.

Our density feature performed well, but computation requires access to the count of training data points that fall inside each polytope. Nevertheless, we argue that these counts could be provided in a privacy preserving manner. For example, $k$-anonymity Sweeney (2002) could be achieved by only sharing counts for polytopes that contain at least $k$ points. When an input lies within a polytope that doesn't meet this threshold (or if the threshold is met but the density is low), the output could be treated as untrusted. Contrast this to traditional density based methods such as Bishop (1994) that require access to the raw training data.

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

# A APPENDIX

## A.1 DATASETS

- Higgs Boson: The Higgs boson dataset from the UCI Machine Learning Repository[2] is used for classification tasks in particle physics. Generated through Monte Carlo simulations, it consists of approximately 11 million entries, each with 28 features. The target variable is binary: 1 (signal) indicates Higgs boson production, and 0 (background) indicates no Higgs boson production. We pre-process the data using PCA to extract 6 high-level features and train a neural network on a sample of the transformed data. The OOD dataset is generated via appendix A.2.

- Circle: The Sphere dataset is synthetic, 3D, and used for binary classification. It is generated with two concentric spheres. Parameters include **n_samples** (total number of data points), **noise** (adds Gaussian noise), **factor** (scale factor for the inner sphere). Features (X) are 3D points (x, y, z), and labels (y) are binary for points on the outer and inner spheres. Challenges include the three-dimensional nature and noise, and potential class imbalance. The OOD dataset is generated via appendix A.2. The Circle dataset is 2D and generated similarly to the Sphere dataset.

- Skin: The Skin Segmentation dataset from the UCI Machine Learning Repository[3] differentiates between skin and non-skin pixels. It consists of 245,057 instances, each representing a pixel. Features include **RGB Values** (intensity of Red, Green, and Blue components) and **Class Label** (1 for skin and 0 for non-skin). The OOD dataset is generated via appendix A.2.

## A.2 GENERATING THE OOD DATA

Out-of-Distribution (OOD) data generation plays a pivotal role in evaluating the robustness and generalization of machine learning models. By intentionally creating data samples that significantly deviate from the training distribution, OOD data tests models against scenarios not covered during training, highlighting potential weaknesses and biases. Statistically, OOD data introduces distributional shifts. Mathematically, this process explores the model's ability to handle unexpected data, emphasizing the importance of designing systems that are resilient to a wide array of real-world conditions. Thus, generating and analyzing OOD data is crucial for advancing machine learning methodologies.

For the Higgs dataset we created OOD dataset as follows. For **OOD_permu** we use item 1, for **OOD_noise** we use item 3, for **OOD_nonlinear** we use item 4 and for **OOD_rearrange** we use item 2.

For the skin dataset we created OOD dataset as follows. For **OOD_min** we use item 1, for **OOD_inv** we change the sign of the data to negative. For the **Texture** we use item 2, for **OOD_diff_dist** we use item 5.

For the circ dataset we created OOD dataset as follows. For **OOD_permu** we use item 1, for **OOD_noise** we use item 3, for **OOD_nonlinear** we use item 4 and for **OOD_max** we use item 6.

1. **Shuffling Data to Create OOD Samples (OOD_permu)**:

    The features (test_x) in the dataset are shuffled column-wise. This means that within each feature column, the values are randomly rearranged. This process ensures that the original relationship between different features in a sample is disrupted, creating out-of-distribution instances.

---

[2]https://archive.ics.uci.edu/dataset/280/higgs
[3]https://archive.ics.uci.edu/dataset/229/skin+segmentation

Our process of generating OOD data through shuffling the columns is designed such that the marginal distribution for any particular column will be unaffected. This creates a challenging OOD dataset where it is not possible to detect data as OOD based on the distribution of any particular column; i.e. trivial OOD approaches based on detecting outliers in a column will not work.

2. **Color Inversion (OOD_rearrange)**: In this procedure for generating out-of-distribution (OOD) data, we start with the original test data from the Skin dataset, which comprises pixel values represented by three columns corresponding to the Red, Green, and Blue (RGB) color channels.

   To create an OOD dataset, we perform a color inversion transformation on the RGB values. The inversion process is achieved by multiplying the values of these three columns by -1. This mathematical operation effectively inverts the color representation, resulting in new RGB values that are the opposite of the original colors in terms of intensity. For instance, a pixel originally having a high intensity in the red channel will now have a high intensity in the opposite direction, and similarly for the green and blue channels.

   This transformation significantly alters the nature of the pixel data while preserving the dimensional structure of the dataset, thus producing OOD data that is fundamentally different in appearance but retains the same overall format as the original dataset. This approach is commonly used to test the robustness of machine learning models in handling data that deviates from the training distribution.

   This method can be generalized and applied to other datasets, particularly when the columns are permuted to create an out-of-distribution (OOD) dataset.

3. **Generating Out-of-Distribution Data Using Gaussian Noise Perturbation (OOD_noise)**: In this procedure for generating out-of-distribution (OOD) data, we start with an original dataset consisting of multiple features organized across different columns. The goal is to introduce randomness into each feature to create an OOD dataset that diverges significantly from the original data while maintaining the same dimensional structure.

   For each column in the dataset, a noise component is added. This noise is generated from a Gaussian (normal) distribution with a mean of 0 and a standard deviation of 1. Specifically, a noise vector with values drawn from this distribution is independently generated for each column, matching the number of rows in the dataset. The generated noise is then added element-wise to the values within that column.

   By repeating this process across all columns, the entire dataset is subjected to random perturbations, ensuring that every feature experiences unique fluctuations. This transformation alters the original values, producing data that deviates from the initial distribution, thus making it out-of-distribution.

4. **Generating Out-of-Distribution Data Using Sine Transformation and Gaussian Noise Addition (OOD_nonlin)**: In this procedure, the goal is to create out-of-distribution (OOD) data by applying a combination of non-linear transformation and noise addition to the original dataset, thereby producing data that is fundamentally different from the original while retaining the same dimensional structure.

   For each feature (column) in the dataset, we perform the following steps:

   (a) **Non-linear Transformation**: Apply the sine function to every value in the column. This transformation changes the original values into their sine equivalents, introducing non-linearity and altering the data's nature. Mathematically, if $x$ represents the original data values in a column, the transformation becomes:

   $$x' = \sin(x)$$

   (b) **Adding Gaussian Noise**: After applying the sine transformation, we generate a noise vector $\epsilon$ from a Gaussian (normal) distribution with a mean of 0 and a standard deviation of 1, matching the number of rows in the dataset. This noise vector is then added element-wise to the transformed values within the column. The resulting transformed data for each feature becomes:

   $$x_{\text{ood}} = \sin(x) + \epsilon$$

   where $\epsilon \sim \mathcal{N}(0, 1)$.

By repeating this process for each column, we obtain a dataset that is both non-linearly transformed and randomly perturbed, resulting in OOD data that is significantly different from the original dataset while maintaining the same overall dimensional structure.

5. **Out-of-Distribution Data Generation Using Feature Transformation and Scaling** This method generates out-of-distribution (OOD) data by transforming each feature of the original dataset to create significantly different distributions. First, the initial feature (representing the 'B' channel) is set to a constant value of 0.5, eliminating any variability. The second feature ('G' channel) undergoes a non-linear transformation by taking the square root of each value, and altering its distribution. Lastly, the third feature ('R' channel) is transformed using a logarithmic function with a small constant added to avoid issues with log(0), which compresses the feature's range. These combined transformations produce an OOD dataset that maintains the original dimensional structure but exhibits fundamentally different characteristics from the original data.

6. **Retaining Maximum/Minimum Feature Values** Let $X_{\text{test}} \in \mathbb{R}^{n \times m}$ be the original test dataset, where $n$ is the number of samples and $m$ is the number of features in each sample. The goal is to create a new matrix $X' \in \mathbb{R}^{n \times m}$ such that for each sample (row) in $X_{\text{test}}$, only the maximum value is retained while all other values are set to zero.

### Mathematical Steps:

1. **Initialization**:

$$X' = \mathbf{0} \quad \text{(an } n \times m \text{ matrix initialized with zeros)}$$

2. **Finding the maximum value index**: For each row $i$ where $i \in \{1, 2, \ldots, n\}$, find the column index $j = \arg\max(X_{\text{test}}[i, :])$ such that

$$X_{\text{test}}[i, j] = \max(X_{\text{test}}[i, k]) \quad \forall k \in \{1, 2, \ldots, m\}$$

3. **Updating the matrix**: Set $X'[i, j] = X_{\text{test}}[i, j]$, where $j$ is the index of the maximum value found in the previous step. All other elements in the row remain zero.

### Final Formulation:

$$X'[i, k] = \begin{cases} X_{\text{test}}[i, k] & \text{if } k = \arg\max(X_{\text{test}}[i, :]) \\ 0 & \text{otherwise} \end{cases}$$

The modified data $X'$ is then saved together with the original labels $y_{\text{test}}$.

### A.3 Universal approximation theorem

We have demonstrated that our method is effective for neural networks with ReLU activation functions. Moreover, our approach is not far from being general. According to the universal approximation theorem Leshno et al. (1993), any continuous function over a compact set can be approximated by a multilayer neural network with ReLU activation. Therefore, even if we use a different activation function, the same approximation can be achieved using a ReLU-based network. With this in mind, we can conclude that our approach possesses a high degree of generality.

### A.4 Evaluation Metrics

In our evaluation, we compare the performance of various methods using several metrics to ensure a comprehensive analysis. One of the primary metrics we utilize is the False Positive Rate at 95% True Positive Rate (FPR95). This metric specifically measures the false positive rate (FPR) of out-of-distribution (OOD) samples when 95% of in-distribution (ID) samples are correctly classified. By focusing on this threshold, we can assess the robustness of different techniques in distinguishing between ID and OOD data.

For the skin dataset we created OOD dataset as follows. For **OOD_permu** we use item 1, for **OOD_inv** we use item 2, for **OOD_diff_dist** we use item 5.

| Dataset | Layer (with Input Dimension) | Number of Neurons | Activation Function |
|---|---|---|---|
| Higgs Dataset | Input Layer (input dim: 6) | 16 | ReLU |
| | Hidden Layer | 16 | ReLU |
| | Output Layer (output dim: 1) | 1 | Sigmoid |
| Skin Dataset | Input Layer (input dim: 3) | 16 | ReLU |
| | Hidden Layer | 16 | ReLU |
| | Output Layer (output dim: 1) | 1 | Sigmoid |
| Circle Dataset | Hidden Layer (input dim: 2) | 12 | ReLU |
| | Output Layer (output dim: 1) | 1 | Sigmoid |

Table 4: Neural Network Models for Different Datasets

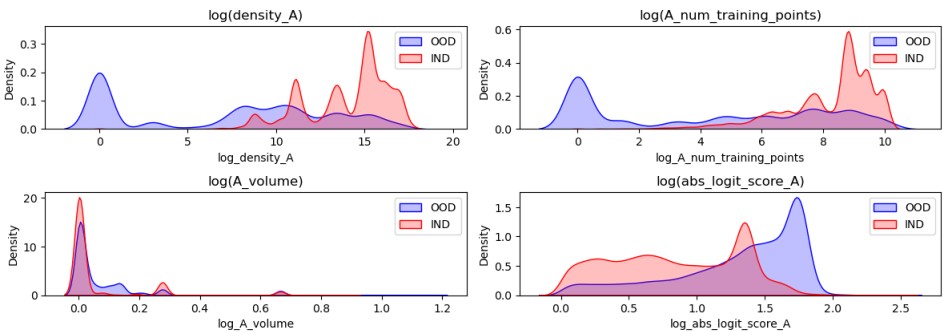

Figure 4: Comparative Analysis of Various Attributes for ID and OOD (shuffle method described in appendix item 1) for the Higgs boson dataset

## B  MODELS USE FOR TRAINING

We give a brief description of the models used for training the neural networks on different datasets.

- For the **Higgs** and **Skin** datasets, the models were compiled using binary cross-entropy as the loss function, Adam optimizer, and accuracy as the performance metric.
- For the **Circle** dataset, the model was compiled with binary cross-entropy loss, Adam optimizer (learning rate = 0.001), and accuracy as the performance metric.

For training the neural network with the Circle dataset, we use 2000 training points and 400 testing points. Additionally, for the out-of-distribution (OOD) dataset, we use 400 points.

For training the neural network with the Skin dataset, we use 245,056 training points. For testing, we use 14,704 points, and for the OOD dataset, we also use 14,704 points.

For the Higgs dataset, we use 200,000 training points and 50,000 testing points. The out-of-distribution datasets consist of the following:

- **OOD_permu:** 50,000 points,
- **OOD_noise:** 2,000 points,
- **OOD_nonlinear:** 20,000 points,
- **OOD_rearrange:** 50,000 points.

