# OpenReview forum: "Tropical Geometry Features for Novelty Detection and interpretability"
_ICLR.cc/2025/Conference — Submitted to ICLR 2025_

### Official Review · Reviewer_EVob · 2024-10-30

**Soundness:** 1
**Presentation:** 1
**Contribution:** 2
**Rating:** 3
**Confidence:** 4

**Summary:**

The paper leverages the piecewise linear structure of ReLU neural networks to draw OOD metrics based on the characteristics of the polytope into which a new test point falls. The metrics are called Tropical Geometry Features (TGF), and are the volume of the polytope, the density of training points in the polytope, and the number of training points in the polytope.

**Strengths:**

Using the decomposition into polytopes for OOD is original and novel.

**Weaknesses:**

However, the paper contains critical flaws:
- The writing is poor. As examples:
  - The term Novelty detection is only used in the title, and then the topic falls back to OOD detection, which is not the exact same topic.
  - In the abstract, l.12 the sentence is not logical: intermediate layers are mentioned and then logits and Energy, whereas these metrics do not use intermediate layers.
  - The introduction is overly long, and some paragraphs sound redundant (ex l.37 - 50).
  - L.104 sentence very vague
  - Notations l.133 - 140 seem not standard, and use standard notations for something else that they are usually used. (ex. \rightarrow)
  - some typos like l.170 stepwise (you meants piecewise I think) l.185 devision
- The paper claims to present a method that improves OOD detection, fairness, adversarial robustness, and uncertainty estimation, whereas the method / experiments are only about OOD detection. A short paragraph is given about how the method might maybe prove useful in the other fields, but claiming it as a contribution is dishonest (it is even in the abstract).
- The benchmark is uncommon for OOD detection, very limited, and only one trivial baseline is used for comparison, even if another baseline is mentioned **in the abstract** (it would be far from enough to include it in a revised version, modern OOD benchmarks are way more furnished). The authors only apply it to binary classification.
- The method is not adapted to high dimensions (the authors admit it but do not go further, even if since we are looking at neural networks we are actually in pretty high dimension) actually, handling polytopes is a NP hard problem for ReLU neural nets [1].
- The presented method is not reproducible. The authors do not explain clearly how the points are mapped to polytopes (I had to guess that each point is mapped to a binary vector corresponding to activation patterns but it is not clearly stated)
- I have concerns about the soundness of the method. Indeed, the number of possible combinations of activation patterns in deep neural nets is exponential ($2^{h_1 + \ldots +h_L}$) so is the number of polytopes. In this context, I do not see how metrics such as density of points or number of points in polytope could work (most of which should be 0).

[1] Jordan, M., & Dimakis, A. G. (2020). Exactly computing the local lipschitz constant of relu networks. Advances in Neural Information Processing Systems, 33, 7344-7353.

**Questions:**

l. 308 "This method requires an effective decomposition of P into simplices, which can be complex, especially
in higher dimensions." how do you cope with this difficulty?

---

### Official Review · Reviewer_imgi · 2024-10-30

**Soundness:** 3
**Presentation:** 2
**Contribution:** 3
**Rating:** 5
**Confidence:** 3

**Summary:**

The paper introduces a novel approach for enhancing OOD detection in neural network by leveraging polytopes derived from learned representation of a network to establish a unique signature of in-distribution data points. The presented method for decomposition of feature vectors is insightful. By analyzing the entire network structure, they create features such as volume and density metrics that offer a more comprehensive understanding of the network's behavior

**Strengths:**

- This paper is well written and structured.
- The chosen approach to resolves an important problem (robustness and reliability)
- Considering features of polytopes like binary vector, volume and density assignment are well considered for improving the robustness and fairness of neural networks models.

**Weaknesses:**

- The experiments were poorly done, and it is recommended that the authors should exhibit results on at least basics OOD or anomaly detection literature benchmarks like Cifar10 - SVHN.
- It is not clear that the proposed method is consistent with scalability of neural networks for bigger, wider and higher dimensional inputs (higher resolution like 224 by 224-pixel images).
- Due to the importance of replication of results for development of research community it is important that any implementation became publicly available. It is recommended that authors use "anonymous for open science" (anonymous GitHub) for their implementation.

**Questions:**

1- Thank you for your submission and well written paper. Despite the goodness of the idea by using polytopes as an approach to solve the important issue of robustness in OOD detection, you did not provide any proven sign that the proposed method will works on CNNs and it is barely mentioned. Which this type of models is the fundamentals of mentioned use cases in paper like "OOD detection", "model interpretability", "adversarial attacks". I would like the authors defend their statements on CNN likes models.

2- In paper section 4.3 authors said: "Note that our goal is not to train a state of the art neural network, but rather
 to provide a sample of ReLU neural networks trained for different datasets on which to test TGF" which is a fair argument for prove of concept, but it is not sufficient for community to trust and build on your work because today's researchers are using very deep and powerful models like ViTs to solve the anomaly and OOD detection problems. Which compel researchers to test they proposed method on at least model like ResNet-18.

3- About the models like ResNet-18 that using batch normalization layer considered as standard technique to achieving higher accuracy. Because of the non-linearity of essence for normalization, proposed method will not be effective. I would like the authors opinion regarding the mentioned problem.

---

### Official Review · Reviewer_FsVr · 2024-11-04

**Soundness:** 2
**Presentation:** 3
**Contribution:** 1
**Rating:** 3
**Confidence:** 4

**Summary:**

The authors present a method to use properties of the polytope where the input data resides for OOD classification. They evaluate various different datasets with various out-of-distribution domains.

**Strengths:**

This paper is clearly written and addresses an important topic: the identification of out-of-distribution points. To my knowledge, the method is novel, and is evaluated on multiple datasets.

**Weaknesses:**

Experiments:
* The main weakness of this paper is that the evaluation setting is too limited. The method is only evaluated on three different datasets which are 6, 3, and 2 dimensional. In order to be a method that is applicable to the broader ML community, evaluations would also need to be conducted on larger scale and more complex datasets. For example, image and natural language datasets such as MNIST, CIFAR, or GLUE, could be reasonable starting places to expand the results to larger (but still simple) datasets. In addition, the use of these larger-scale and higher-dimensional datasets will allow the use of more modern machine learning architectures & models which may make the results of the study more applicable to the community.
* The authors should include comparisons to prior work. The methods evaluated in [1] would be a good starting place, but there is a relatively large body of relevant OOD detection literature that the authors should explore, for example as described in this survey [2].

[1] Tajwar, Fahim, et al. "No true state-of-the-art? ood detection methods are inconsistent across datasets." arXiv preprint arXiv:2109.05554 (2021).
[2] Yang, Jingkang, et al. "Generalized out-of-distribution detection: A survey." International Journal of Computer Vision (2024): 1-28.

**Questions:**

See weaknesses.

---

### Comment · Area_Chair_sGov · 2024-11-22

Dear Authors and Reviewers,

The discussion phase has passed 10 days. If you want to discuss this with each other, please post your thoughts by adding official comments.

Thanks for your efforts and contributions to ICLR 2025.

Best regards,

Your Area Chair

---

### Meta-Review · Area_Chair_sGov · 2024-12-20

**Metareview:**

This paper argued that previous work overlook the geometric properties of the learned representations in the latent space, failing to capture important signals that relate to model reliability, fairness, and adversarial vulnerability. Then, the authors proposed Tropical Geometry Features (TGF) for detecting out-of-distribution data and enhancing overall model evaluation. The point made by this paper is interesting. However, the empirical evaluation is too limited, making this paper have a limited contribution to the field.

**Additional Comments On Reviewer Discussion:**

No rebuttal is provided by the authors.

---

### Decision · Program_Chairs · 2025-01-22

Reject